# Defect Structures of Magnetic Nanoparticles in Smectic A Liquid Crystals

**DOI:** 10.3390/molecules26185717

**Published:** 2021-09-21

**Authors:** Vladimíra Novotná, Lubor Lejček, Věra Hamplová, Jana Vejpravová

**Affiliations:** 1Institute of Physics of the Czech Academy of Sciences, Na Slovance 2, 182 21 Prague, Czech Republic; lejcekl@fzu.cz (L.L.); hamplova@fzu.cz (V.H.); 2Department of Condensed Matter Physics, Faculty of Mathematics and Physics, Charles University, Ke Karlovu 5, 121 16 Prague, Czech Republic; jana@mag.mff.cuni.cz

**Keywords:** liquid crystals, smectics, dislocation loops, inclusions

## Abstract

Topological defects in anisotropic fluids like liquid crystals serve as a playground for the research of various effects. In this study, we concentrated on a hybrid system of chiral rod-like molecules doped by magnetic nanoparticles. In textures of the smectic A phase, we observed linear defects and found that clusters of nanoparticles promote nucleation of smectic layer defects just at the phase transition from the isotropic to the smectic A (SmA) phase. In different geometries, we studied and analysed creation of defects which can be explained by attractive elastic forces between nanoparticles in the SmA phase. On cooling the studied hybrid system, clusters grow up to the critical dimension, and the smectic texture is stabilised. The presented effects are theoretically described and explained if we consider the elastic interaction of two point defects and stabilisation of prismatic dislocation loops due to the presence of nanoparticles.

## 1. Introduction

Liquid crystals (LCs) represent self-assembled media possessing an orientational order of molecules as well as fluidity inherent to liquids [1]. Comprising molecules can form a variety of mesophases, which are very susceptible to external stimuli and influences. If LCs are subjected to specific boundary conditions, point, planar, or plane defects can appear. Generally, defects are not desirable as they lower the performance of LC applications. On the other hand, topological defects are the subject of sophisticated engineering. A regular liquid crystalline structure can be disrupted by admixed nanoparticles (NPs), which interact with topological defects, and NPs can be attracted to them. Generally, the interaction between particles and LC molecules depends on nanoparticle size, shape, and physical properties of both partners, NPs and organic molecules. Within the last two decades, the systems of LCs and nanoparticles have been intensively studied [2] and it has been confirmed that the orientational and positional ordering of mesogenic molecules can induce unexpected effects in the presence of nanoparticles [2,3,4,5,6,7,8].

When particles are dispersed in a nematic (N) liquid crystal, the orientation of N molecules is locally disturbed due to their interaction with inclusions. Although anisotropic dispersions of NPs in nematics have been pursued repeatedly [9,10,11,12], admixed nanoparticles have not been studied intensively in smectic phases [13,14,15,16,17,18]. Let us concentrate on smectic A (SmA) phases, in which molecules are packed in layers in a parallel direction to the layer normal, and smectic C phases (SmC) with tilted molecules with respect to the layer normal. Point defects in the layered structure of smectic LCs lead to volume layer deformations, and elastic properties play an important role. Layer deformations are usually accompanied by perturbations of molecular director orientations (see, e.g., [15]). Layer deformations near point-like inclusions were theoretically studied and described in several papers [16,17,18,19]. Another type of defects has been observed in thin free-standing smectic films, where inclusions penetrate through the whole film. The molecular anchoring on the surface of inclusion leads to the creation of disclinations and, therefore, to long-range interactions between inclusions. There are many studies, including review articles [20,21,22,23,24], concerning this behaviour. Nevertheless, so far, there have been only a few reports on hybrid systems of smectic liquid crystals and magnetic nanoparticles, especially on the formation of defects and their understanding [2,25,26]. Computer simulations based on the Monte–Carlo method were presented [27,28] and predicted possible scenarios for ordering and phase organisations.

In this article, we present our observations of a smectic compound admixed with a small-weight quantity of magnetic nanoparticles. We studied the hybrid system under a polarising light microscope in various geometries as confinement at different surfaces influences the studied defect structure. We found that there are nanoparticle clusters arising at the isotropic–smectic A phase transition on cooling the hybrid system. Under a polarising microscope, we observed a growth of nuclei on cooling from the isotropic phase, and textural features persisted deeply into the SmA phase. We aimed to explain our observations using a theoretical description based on elastic interactions of point defects.

## 2. Preparation of the Hybrid System

As the host for the preparation of a hybrid system, we utilised the thermotropic liquid crystalline compound designated DL12/*. Previously, this type of compound was synthesised in our laboratory, and mesomorphic properties were studied [29]. The molecular core of the DL series consisted of three phenyls connected by ester linkage groups. Two chiral centers were applied in the chiral chain, namely methylbutyl and lactate moiety [29]. The host compound DL12/* exhibits the smectic A (SmA) phase below the isotropic (Iso) phase temperature on cooling the sample. The temperature of the Iso–SmA phase transition, T_iso_, is about 100 °C, and the SmA–SmC* phase transition was found at T_C_ = 79 °C. The ferroelectric SmC* phase persists down to room temperature during the cooling process; then, crystallisation occurs. On heating, the melting point (m.p.) was observed at 48 °C. The layer thickness, *d*, for DL12/* was established based on X-ray measurements as 4 nm [29].

In the next step, we admixed DL12/* with cobalt-ferrite (CoFe_2_O_4_) magnetic nanoparticles (MNPs). We used a hexane dispersion of cobalt-ferrite MNPs of spherical shape with a diameter of 6–8 nm, coated with oleic acid. The CoFe_2_O_4_ MNPs were synthesised by hydrothermal decomposition of iron nitrate and cobalt nitrate in the presence of sodium oleate reported previously [30]. The hybrid system with the concentration of 1 weight % of CoFe_2_O_4_ and 99 weight % of DL12/* was prepared and dissolved in 5 ml of hexane (p.a.) under heating and stirring. Relative concentrations of components were adjusted to obtain 20 mg of the final composite. The hexane solution was then heated to 60 °C, and the solvent was gradually evaporated. The composite was then dried in a vacuum oven for two hours at room temperature.

We studied the hybrid system with 99% *w*/*w* of DL12/* and 1% *w*/*w* of CoFe_2_O_4_ and found that mesomorphic properties were changed compared to the host compound DL12/*. The phase transition temperatures T_iso_ and T_c_ were shifted by 7 °C; at T_iso_, coexistence of the isotropic and SmA phases was observed between 93 °C and 90 °C on cooling the hybrid system from the isotropic phase. For the observation under a polarising microscope, several types of samples were prepared in different geometries. We utilised various kinds of cells, a home-made cell without a surfactant and a commercial cell (both with the thickness of 5 μm), with glass surfaces provided with ITO electrodes. In the commercial cell, a surfactant ensures homogeneous alignment (HG) of molecules parallel to surfaces. The cells were filled with a studied hybrid composite by capillarity in the isotropic phase. Other type of samples was prepared in the form of a droplet spread on a glass plate, which was without any surfactant or surface treatment.

## 3. Results

### 3.1. Defects Observed in Polarised Light

We pursued the above described hybrid system of liquid crystalline compound DL12/* with MNPs in different cell geometries under a polarising optical microscope. At first, the hybrid sample with nanoparticles was studied in home-made cells, which were prepared from glass without any alignment at surfaces. We observed numerous objects having the Maltese cross-like contrast (Figure 1 and Figure 2) below the Iso–SmA phase transition temperature. In such cells, we can expect natural preferences of the molecular orientation. The observed nuclei were homogeneously generated with very narrow distribution of diameters from 11.5 μm to 15 μm, as shown in Figure 1. The average distance between object centres was 12–16 μm, which means that there was a gap of about 0.5–1 μm between the nuclei. A quasi-hexagonal packing was present when the concentration of nuclei was high (Figure 1). The observed objects can be interpreted as layer defects in the SmA phase, like small focal conics or small prismatic dislocation loops.

Additionally, commercial HG cells with homogeneous anchoring of molecules (parallel to surfaces) were utilised. Nevertheless, due to positive dielectric anisotropy of the studied hybrid system the applied DC electric field caused reorientation of molecules along the applied field, and the textures were modified in the SmA phase. After switching off the field, we were able to study growth of the nuclei. In the bottom left corner of Figure 2a, we demonstrate how the area under electrodes was transformed by the applied electric field. In Figure 2b, after switching off the electric field, the nuclei were observed. They can be visualised in the depolarised view (Figure 3b) when the contrast of defects is sharper. For comparison, in the pure liquid crystalline compound DL12/*, after switching off the applied field, we observed batonnet-like nuclei with quicker recovery to the fan-shaped texture. Unfortunately, direct application of commercial cells with homeotropic anchoring (HT cells) did not supply textures with clear structural features and unambiguous differences between pure DL12/* and the hybrid system with MNPs.

For comparison, in the sample of pure liquid crystalline compound DL12/* without nanoparticles, the situation differs from the studied hybrid system. For pure DL12/* compound in the cell without a surfactant, the population of nuclei is not regular, and the nuclei prefer to grow instead of generating new ones at the isotropic–SmA phase transition. The nuclei can disappear as they are not stable, and the condition for their instability is defined by line tension. We can compare these two samples and conclude that the observed nuclei are regularly populated in the hybrid system with nanoparticles (Figure 1). They are stable within a broader temperature range, and the precipitation of nanoparticles at these defects can be supposed. Previously, we observed inclusions at silver nanoparticles in the SmA phase in a similar liquid crystalline system and concluded that the presence of nanoparticles leads to layer deformation and creates a tilt in their vicinity, which can be observed under a polarising microscope [15]. A similar effect could be expected in the hybrid system in this study, and we describe it in the following paragraph.

A detailed study was devoted to droplets with one free surface as the geometry of such a sample is more complicated. There are various molecular orientations at different borders (glass plate–hybrid compound and compound–air at the free surface), and the thickness of the sample changes as well. Such a droplet for the studied hybrid system with nanoparticles is presented in Figure 4. Figure 4a shows the view in polarised light under the crossed position of polarisers, while Figure 4b was taken in depolarised light. In thicker parts of the droplet, we again observed a system of toroidal focal conics, on which inclusions are generated. On the other hand, near the sample’s edge, one can see stripes visible both in polarised and nonpolarised light. It is difficult to identify the nature of those stripes exactly. We can speculate that they are due to the edge dislocations with a greater Burgers vector formed at the sample edge where the thickness changes. It is necessary to point out that such stripes were not observed in the pure liquid crystalline sample without nanoparticles. Focal conics are toroidal domains, and they were observed in the thicker part of the sample with one free surface. The focal lines can be visualised as circles, while the other line is projected as a dot if it is perpendicular to the plane of Figure 4. The radius of the nuclei proportionally grew with cell thickness. For cell thickness larger than about 10 μm, nonhomogeneous distribution of the radius was observed (see Figure 4c).

### 3.2. Model and Discussion

In this section, we propose a model based on our observations which permits us to describe the studied hybrid system. First, let us note that MNPs are covered by an organic layer from oleic acid, which protects nanoparticles from a direct contact. However, we suppose that agglomeration can appear during thermal treatment of a hybrid sample within a liquid, as well as the LC phase existence. Due to the influence of elastic forces, we assume nanoparticles can freely move by diffusion in the liquid crystalline phase and form clusters. As the clusters of nanoparticles have dimensions of micrometres, one can expect that they are structured into magnetically interacting aggregates. However, at room temperature and without the external magnetic field, we assume that the orientation of the MNPs’ macrospins within an inclusion will be random, thus macroscopically compensated. This is corroborated by the fact that individual MNPs in aggregates, despite a close contact and strong interparticle interactions, are in the superparamagnetic regime as their blocking temperature is far below room temperature [30]. The resulting magnetostatic forces will be negligible, and so we can describe the formation of clusters on the basis of smectic elasticity.

To discuss the formation of inclusions in the smectic A phase, we shortly reviewed the elastic interaction of two point defects. The definition of the coordinate system in the smectic A phase is shown in Figure 5. The *z*-axis is along with the layer normal, while the plane (*x,y*) is parallel to the smectic A planes. We know that the layer thickness, *b*, for DL12/* is about 4 nm and the size of MNPs is 6–8 nm. Considering the coating of nanoparticles with oleic acid, an inclusion of MNPs includes at least three layers.

The displacement of smectic layers caused by point defects was previously described [16,17,18]. The layer displacement of smectic layers in the *z*-direction caused by a point force acting also in the *z*-direction was given by Green’s function of smectic A elasticity. The interaction of two point defects was investigated by Terentjev [18] and also by Turner and Sens [19]. Let the first point defect be centred at the coordinate origin and the centre of the second point defect be situated at point $r_{o}=\left(x_{o},y_{o},z_{o}\right)$. If the inclusion volumes are $\delta V_{1}$ and $\delta V_{2}$ and they are supposed to be minor, the interaction energy $\phi_{int}$ of both point defects was determined by Terentjev [18] as follows:(1)$$\phi_{int}=-\frac{B\delta V_{1}\delta V_{2}}{8\pi \lambda }\frac{1}{z_{o}^{2}}e^{-\frac{\rho_{o}^{2}}{4\lambda \left|z_{o}\right|}}\left(1-\frac{\rho_{o}^{2}}{4\lambda \left|z_{o}\right|}\right)$$
where $\lambda =\sqrt{\frac{K}{B}}$ is the de Gennes length, K is the elastic modulus of the curvature, and B characterises the elastic volume compressibility of smectic layers. The distance between point defect centres in the plane of smectic layers is $\rho_{o}=\sqrt{x_{o}^{2}+y_{o}^{2}}$. The parameter Ω, which was defined in [18], can be related to the inclusion volume δV as Ω=B δV. The radial force between two point defects, i.e., the force acting parallel to smectic planes, is $f_{\varrho }=-\frac{\partial \phi_{int}}{\partial \rho_{o}}$. Then, the radial force component is as follows:(2)$$f_{\varrho }=-\frac{B\delta V_{1}\delta V_{2}}{8\pi \lambda^{2}}\frac{\rho_{o}}{{\left|z_{o}\right|}^{3}}e^{-\frac{\rho_{o}^{2}}{4\lambda \left|z_{o}\right|}}\left(1-\frac{\rho_{o}^{2}}{8\lambda \left|z_{o}\right|}\right)$$
and the *z*-component of the force oriented along the normal to layers is as follows:(3)$$f_{z}=-\frac{B\delta V_{1}\delta V_{2}}{4\pi \lambda {\left|z_{o}\right|}^{3}}e^{-\frac{\rho_{o}^{2}}{4\lambda \left|z_{o}\right|}}\left(1-\frac{\rho_{o}^{2}}{2\lambda \left|z_{o}\right|}+\frac{\rho_{o}^{4}}{32\lambda^{2}z_{o}^{2}}\right)$$

Note that the volume of the point defect δV≈ξδA, where δA is the surface of the point defect in the plane of the smectic layer while the parameter ξ is the dimension of the point defect perpendicular to smectic layers. This parameter is responsible for layer deformation caused by a point defect embedded within a smectic layer or between smectic layers. As discussed by Lejcek [17] and Terentjev [18], δA can be imagined as a surface of an infinitesimal prismatic dislocation loop while ξ can be understood as its Burgers vector along the normal to smectic layers.

We applied these evaluations to discuss the observed phenomena. When a smectic liquid crystalline compound contains nanoparticles, they can move through the system of smectic layers by diffusion very easily, namely, when the temperature is near the transition from the isotropic phase to the smectic A phase. Moreover, just after transition, many small prismatic dislocation loops or small focal conic domains are created (as can be demonstrated in Figure 1). In the center of the dislocation loop, there is vacant volume. Without nanoparticles, this object is energetically unstable. On the other hand, in the presence of nanoparticles, this vacant volume can be attractive for them. Dislocation components of the dislocation loop interact with nanoparticles and attract them to the volume compression side of dislocation [17]. The vacant volume, schematically drawn in Figure 5, corresponds to such a compression volume of the prismatic dislocation loop.

The observed defects can serve as centers for the nucleation of nanoparticle clusters. When a cluster is nucleated, it attracts other point defects as observed in force Equations (2) and (3). At temperatures just below the phase transition, the volume compression modulus *B* is small. The force $f=\left(f_{\varrho },f_{z}\right)$ between nanoparticles assists the diffusion motion of further nanoparticles to a nucleus of the cluster within smectic layers. For this reason, below the Iso–SmA phase transition, the clusters tend to grow. Parameter $\varrho_{o}$ limits the area from which nanoparticles are attracted to the cluster. In the domain limited by the radius $\varrho_{o}<2\sqrt{2\lambda \left|z_{o}\right|}\;\;$ the force $f_{\varrho }<0$ is attractive, and nanoparticles within this domain tend to move to the nanoparticle cluster, while outside this domain, the point defects are repulsed from the cluster. At temperatures near the transition, λ has high values [31,32], so $\rho_{o}$ can also be high_._ With the decrease in temperature further from the phase transition, λ also decreases, and the domain limited by $\rho_{o}$ also decreases. When the radius of the inclusion $a_{c}$ reaches the critical value $\rho_{o}$ at a given temperature, i.e., $a_{c}\approx \;\rho_{o}$, the growth of the inclusion is terminated as other nanoparticles outside this radius are repulsed. Therefore, when an inclusion reaches the critical radius $a_{c}$ in the plane of smectic layers, its growth in this plane is finished. The size of the inclusion in the *z*-dimension depends on the sign of the $f_{z}$ component.

Figure 6 shows a schematic model of prismatic dislocation loops around a nanoparticle cluster. The system is stable as dislocation loops cannot collapse by line tension. The clustering of nanoparticles is natural, and clusters penetrate several neighbouring smectic layers. The radius of nanoparticles (6–8 nm) is greater than the smectic layer thickness (4 nm). Additionally, the oleic acid presence causes the radius of nanoparticles effectively to be even higher. Nevertheless, the individual prismatic loops are not distinguishable under an optical microscope. The small prismatic loops group together to finally form a single loop of giant dislocation surrounding the cluster of nanoparticles. We assume that clusters were more elongated in the direction of the layer normal (*z*-direction). This direction corresponds to the axis of cylindrical symmetry for the SmA phase. It is consistent with our observation under a polarising microscope. When we focused on the upper or lower surface of the sample, the diameter of the nuclei did not change.

Point defects forming the inclusion increase the inclusion volume. However, it should be noted that force Equations (2) and (3) are valid for point defects of small volume, not for inclusions with volumes of the micrometre radius. The clustering of point defects affects many smectic layers (Figure 6) and the elastic deformation is then provided by an array of dislocation loops. It should be noted that the deformation of layers also locally influences the orientation of molecules around the cluster. The interaction of inclusions can be described as the interaction of prismatic dislocation loops. This interaction is not strong, and it does not influence significantly the inclusion configuration created just after the phase transition. The mobility of inclusions that reached the critical radius is therefore practically zero. This fact also stabilises the inclusion configuration. We observed that the process of clustering was thermally reversible. After the transition to the isotropic state, most of the clusters dissolved, and when the temperature decreased again, the process of cluster formation was reproduced.

## 4. Conclusions

We investigated textures of a liquid crystalline compound doped with cobalt-ferrite MNPs at the isotropic–SmA phase transition and in the SmA phase. The presence of nanoparticles induces broad-range coexistence of the SmA and isotropic phases, which causes stabilisation of the regular defect system. Due to the self-assembling properties of liquid crystals, the inclusions can interact at a larger distance than the nanoparticle size. We interpret the effects observed by polarised optical microscopy as clustering of nanoparticles on layer defects created below the phase transition temperature on cooling the sample. Based on a comparison with a pure liquid crystalline compound, we can conclude that the presence of nanoparticles makes the defect system stable in a broad temperature range.

We consider the clustering is originated at first by the interaction of nanoparticles with dislocation segments of small prismatic dislocation loops rather than by magnetic forces. We expect that nanoparticles start concentrating in the vacant centers of dislocation loops. Then, nuclei of clusters interact and attract other nanoparticles, which leads to the cluster’s growth. This process is essential just below the phase transition temperature. When the cluster reaches the critical radius, the growth is finished and the texture is stabilised. We propose a model to describe and explain how macroscopic texture reflects microscopic defect structures, as MNPs are confined at generated defects.

## Figures and Tables

**Figure 1 molecules-26-05717-f001:**
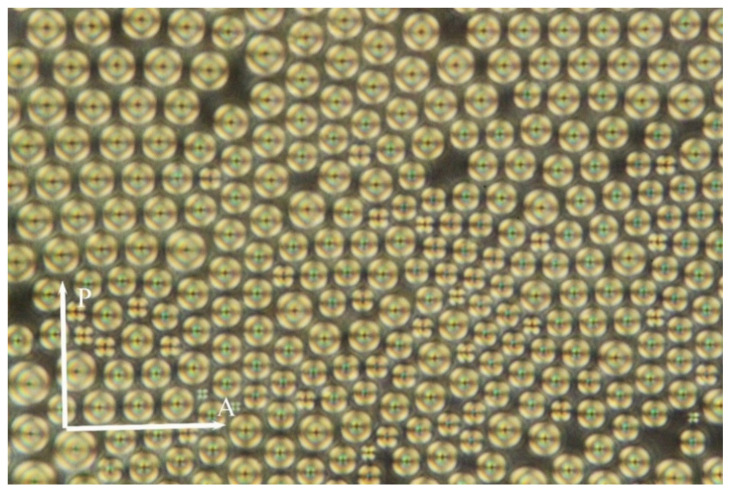
Texture of inclusions in a hybrid system of liquid crystalline compound DL10* with magnetic nanoparticles at 5 °C below the isotropic–SmA phase transition temperature. Orientation of the polariser (P) and the analyser (A) is depicted. The width of the picture is 240 μm.

**Figure 2 molecules-26-05717-f002:**
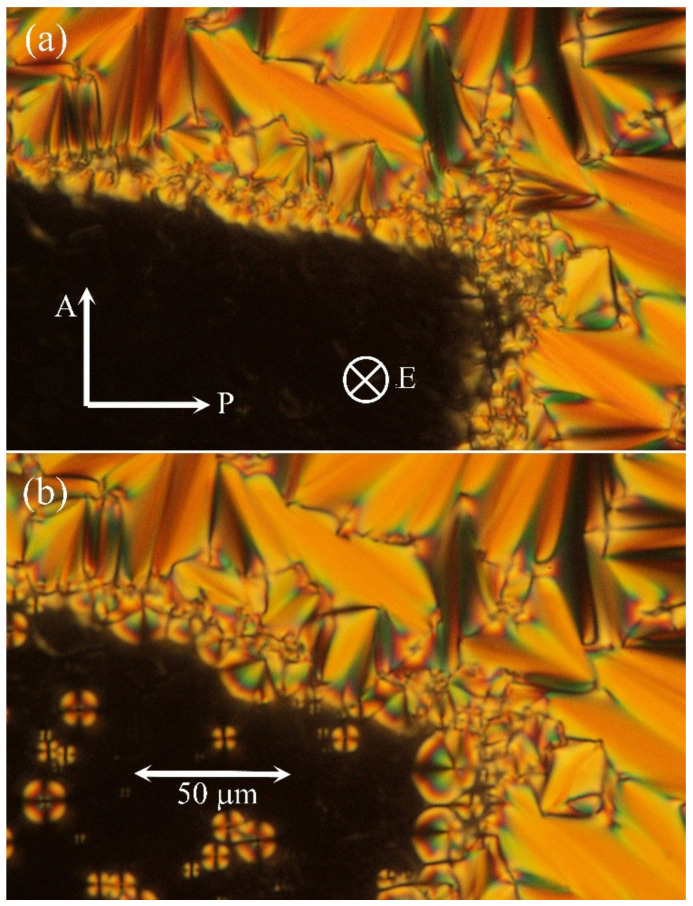
Microphotographs of the studied hybrid system in a commercial HG cell, thickness—5 μm, in the SmA phase at T = 98 °C (**a**) under the applied electric field, E, with the intensity of 50 V/μm in the perpendicular direction, and (**b**) after switching off the field after 10 seconds.

**Figure 3 molecules-26-05717-f003:**
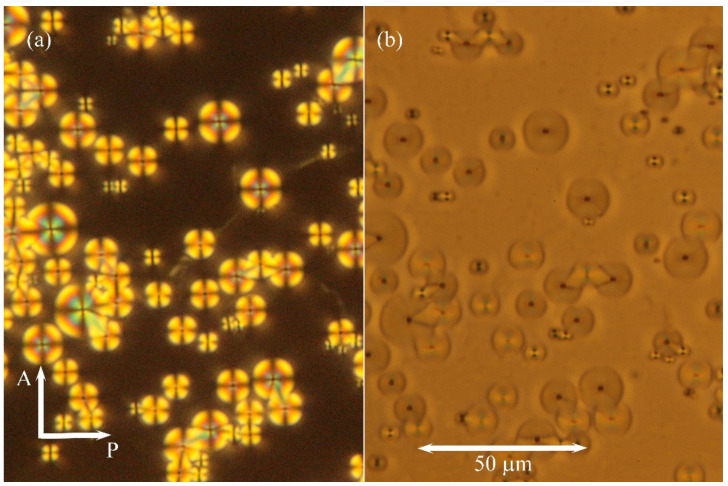
Texture of small focal conics and small prismatic dislocation loops in the studied hybrid composite system: (**a**) in the crossed position of the analyser (A) and the polariser (P) and (**b**) in depolarised light. The width of the picture is 120 μm. Clusters of nanoparticles are visualised as black dots accompanied by line defects, either focal lines or dislocation lines.

**Figure 4 molecules-26-05717-f004:**
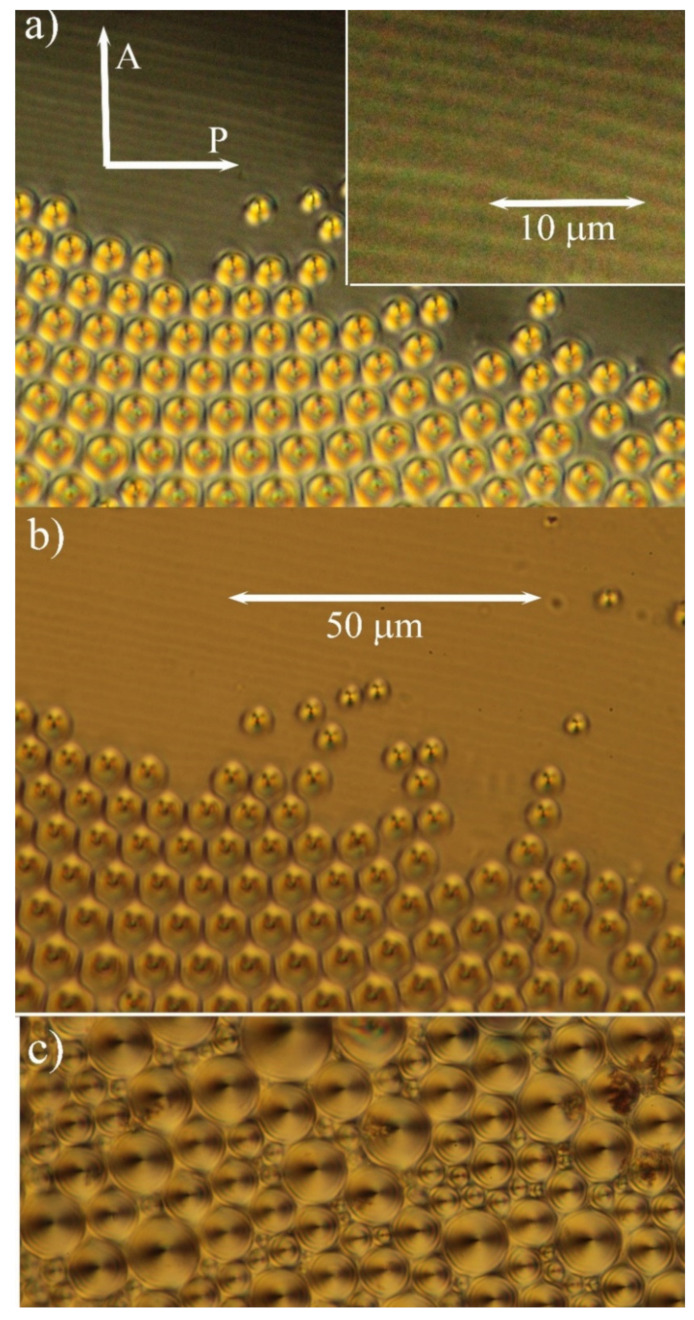
Microphotographs of a droplet on a glass plate: (**a**) under crossed polarisers and (**b**) the identical part of the sample in depolarised view with a common scale bar for all three figures. The inset in (**a**) shows the border of the droplet in an enlarged view and (**c**) presents focal conics in the thicker part of the droplet.

**Figure 5 molecules-26-05717-f005:**
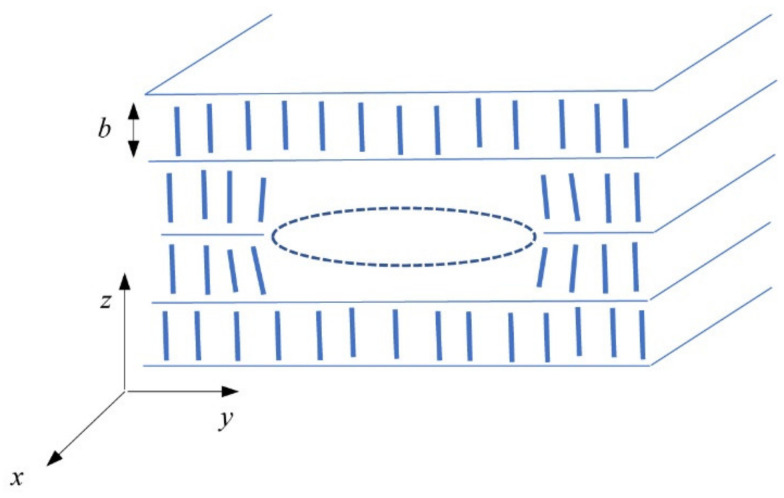
Schematic drawing of a prismatic dislocation loop with the Burgers vector 2*b* in smectic layers. Rod-like molecules in layers are presented as small rods. A vacant volume is designed in the center of the loop where nanoparticles can condense. Orientation of the applied coordinate system is depicted; note that the *z*-axis is oriented along the layer normal.

**Figure 6 molecules-26-05717-f006:**
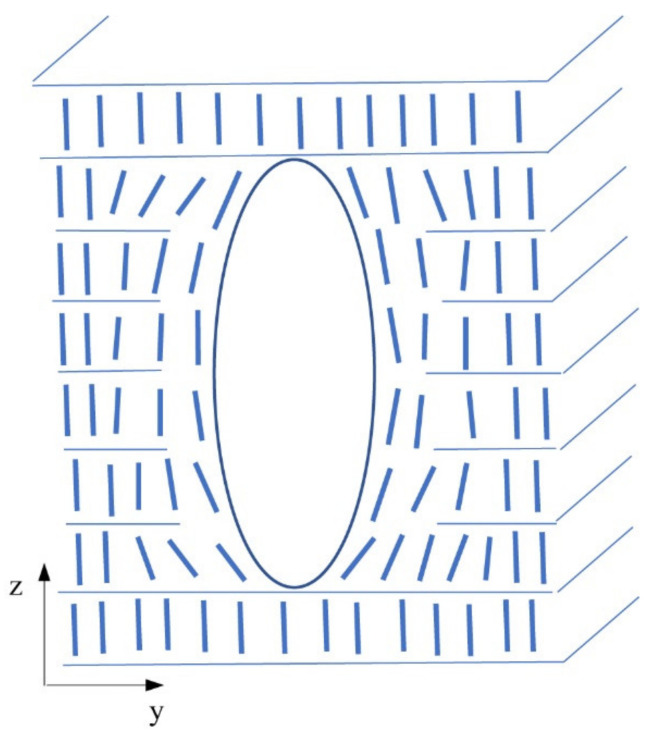
Schematic drawing in the plane (*y*,*z*) shows a nanoparticle cluster incorporated into the smectic layer system. The cluster in this scheme has an ellipsoidal profile along with the layer normal and is covered by a shell of disordered molecules influenced by the molecular interaction with the cluster’s surface. The cluster within the shell is enveloped by prismatic dislocation loops (not included in this figure, for the illustration see Figure 5).

## Data Availability

Not applicable.

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
