# Peer review of "Defect Structures of Magnetic Nanoparticles in Smectic A Liquid Crystals"

_molecules, 2021, doi:10.3390/molecules26185717_

Round 1

Reviewer 1 Report

The present work and idea are not novel. No new findings or conclusions were reported. The novelty of stated/finding results is very limited. The most of conclusions are already reported by several groups around the world. Accordingly, the contribution of the current study to the scientific community will be very poor. Sorry, I can NOT recommend the publication of the present study.

Author Response

Such statements are generic and we disagree. Up to now, there are only limited number of papers about defects on magnetic nanoparticles in smectic A phases. Our observations are original and the studied arrays of defects are unique (Figure 1 and Figure 3). The reviewer should provide more arguments and references, as we do not find any support for his/her conclusion.

Reviewer 2 Report

In this manuscript, the authors have presented a study on the self-assembly of a hybrid mixture of liquid crystals and magnetic nanoparticles. In particular, the work focuses on exploring the defect structure at cobalt-ferrite magnetic nanoparticles doped inside a liquid crystalline matrix of DL 12/*, chiral rod-like molecules. The authors have studied the generation defect structures in various geometries and provided a theoretical explanation based on elastic interactions between the point defects in the smectic phase. The subject of the paper is timely as various liquid crystals and magnetic nanoparticle composites attract much attention and are expected to possess advantageous functional properties. The article is concise, well-written, and scientifically sound. However, I feel that the introduction part should be extended by providing more details on the literature related to the self-assembly of liquid crystal and magnetic nanoparticle systems. In this context, computer simulation works can also be included, e. g., S. D. Peroukidis et al., Soft Matter 11, 5999 (2015), S. D. Peroukidis et al., Phys. Rev. E 92, 010501(R) (2015), G. P. Shrivastav et al. Soft matter 15, 973 (2019), G. P. Shrivastav, Crystals 11, 834 (2021). Furthermore, a summary of different sections can be added at the end of the introduction.

Overall, the manuscript is well organized. However, there are some "spots" throughout the text, which merit editorial amendments.  I recommend the manuscript for publication in Molecules after the authors address the issues mentioned above.

Author Response

We thank the reviewer for his/her kind remarks and recommendations. We added recommended references and tried to improve the text.

Reviewer 3 Report

In my opinion, the submitted manuscript is nicely written  and can be very interesting for t researchers specializing in the field of the dispersion of nanoparticles in LCs. I would advise to accept this manuscript for publication/ 

Author Response

We thank very much to the reviewer for his/her careful reading of manuscript.

Reviewer 4 Report

This paper reports observations on the nucleation of defects (toric domains) in the smectic A phase of a liquid crystal designed as DL12/* doped with magnetic cobalt-ferrite nanoparticles. The latter are functionalized with oleic acid to ensure a good dispersion in the isotropic phase of the liquid crystal. The main observation is that the nanoparticles aggregate at the isotropic-to-smectic A phase transition and form solid particles of typical radius 1 micrometer in the plane of the layers. These aggregates act as nucleation centers for toric domains which are approximately all of the same size (between 12 and 16 micrometers). This behavior is explained in terms of elastic interactions between the nanoparticles mediated by the deformation of the layers.

 These observations are rather interesting and could deserve publication. Nevertheless, I have questions and suggestions the authors should consider before publication.

  • What is the thickness of the homemade cell? Does the size of the toric domains scale with the thickness of the sample in figure 1? Did you try to change the thickness of the sample to test this point?
  • Line 88. What means HG (homeotropic geometry?). When you speak about planar geometry, does it mean that the layers are perpendicular to the ITO electrodes? If it is the case, are the surfaces rubbed in a single direction as it is usually the case in commercial cells? Which surfactant is used? (knowing that with a surfactant a homeotropic anchoring is usually obtained).
  • From the band texture visible in the meniscus, I suspect that this photo has been taken in the smectic C* phase? Am I right. Please specify the temperature.
  • In my opinion, the observations on free standing films do not bring any new insight into the problem of the nucleation of toric domains and could be removed.
  • Line 157. What do you mean when you say that clusters of NP are pushed towards the meniscus due to the action of surface tension?
  • Line 215: de Gennes is misspelled.
  • Lines 270-271. I do not understand how you obtain the formula used to estimate the penetration length l. In addition, the value that is calculated, i.e. 17.6 mm, is much too large in my opinion. By taking K=10-7 dyn, this would give B≈0.03 dyn/cm2. This is not realistic at all, even at the smectic A-to-isotropic phase transition that is first order.  
  • Figure 7 This model is strange. How do you kwow that the aggregates are elongated along the normal to the layers. In addition this model seems optically incompatible with the observations of figure 1.
  • Did you change the concentration of nanoparticles to check whether the size and the density of aggregates change? This could be interesting to develop a more convincing aggregation model.

Author Response

We thank very much to the reviewer for his/her careful reading of manuscript.

Concerning recommended improvement, we tried our best to meet all his/her remarks and suggestions.

We added information about thickness of our home-made cell.

  • Line 88. What means HG (homeotropic geometry?). When you speak about planar geometry, does it mean that the layers are perpendicular to the ITO electrodes? If it is the case, are the surfaces rubbed in a single direction as it is usually the case in commercial cells? Which surfactant is used? (knowing that with a surfactant a homeotropic anchoring is usually obtained).

We modified the text about homogeneous boundary conditions in HG cell.

  • In my opinion, the observations on free standing films do not bring any new insight into the problem of the nucleation of toric domains and could be removed.

We agree that only limited information is delivered concerning free-standing films. So we decided to remove Figure 4 and corresponding part of text.

  • Line 215: de Gennes is misspelled.

We improved the misprint in the spelling for the name in “de Gennes”.

  • Lines 270-271. I do not understand how you obtain the formula used to estimate the penetration length l. In addition, the value that is calculated, i.e. 17.6 mm, is much too large in my opinion. By taking K=10-7dyn, this would give B≈0.03 dyn/cm2. This is not realistic at all, even at the smectic A-to-isotropic phase transition that is first order.  

Concerning our estimation of nuclei radius (line 270-271), we based it on experimental values for another type of LC systems. Then we decided to remove this part from the manuscript. Generally, it is redundant as in our particular case we cannot compare in such detailed analysis with experimental data.

  • Figure 7 This model is strange. How do you know that the aggregates are elongated along the normal to the layers. 

We explained why we suppose that the aggregates are elongated along the normal to the layers.

  • Did you change the concentration of nanoparticles to check whether the size and the density of aggregates change? This could be interesting to develop a more convincing aggregation model.

Concerning the nanoparticle size, we applied the smallest available nanoparticles to fit to nano-scale organization of liquid crystals.

Round 2

Reviewer 4 Report

This paper is now acceptable for publication in Molecules. I nonetheless suggest that the authors correct the following points:

  1. In page 3 it is written “In commercial cell a surfactant ensures homogeneous alignment (HG) of molecules in a planar geometry”. Could the authors precise which surfactant is used? Could the authors also precise what means planar in this context. Does it mean that the molecules are parallel to the surface and all aligned in the same direction? In that case, are the surfaces rubbed in a single direction? Or does it mean that the layers are parallel to the glass plates. In that case I would speak rather about homeotropic alignment of the molecules. What means HG (homeotropic geometry ???).
  2. In page 3 remove this sentence “Additionally, we prepared a free-standing film by spreading the melted material over the hole in a metal plate, ensuring a homogeneous geometry”.
  3. My main problem concerns the model shown in figure 6. It is not compatible with the observations. Indeed, according to the authors, the nanoparticules form elongated clusters of typical diameter 1 micrometer in the plane of the layers. In the drawing, the layers are curved within a domain of diameter barely larger than the one of the cluster which is incompatible with the observations between crossed polarizers. Could the authors discuss this point in more detail? Does it mean that the small prismatic loops group together to finally form a single loop of giant dislocation surrounding the cluster of nanoparticles? This could explain better the observations in my opinion.

Author Response

  1. We tried to improve the text about cells to describe the geometry.

  2.  

    We remove the text, which is redundant as we deleted the part relevant to free-standing films.

  3.  

    We modified Figure 5 and Figure 6. The referee is right that the small prismatic loops group together to finally form a single loop of giant dislocation surrounding the cluster of nanoparticles.

This manuscript is a resubmission of an earlier submission. The following is a list of the peer review reports and author responses from that submission.

Round 1

Reviewer 1 Report

Referee report on # 1288094

submitted for publication at Nanomaterials

Title: Defect structure at magnetic nanoparticles in smectic A liquid crystals

By: Vladimira Novotna, Lubor Lejcek, Vera Hamplova, Jana Vejpravova

This paper reports on experimental investigations of topological defects in Smectic A liquid crystal associated to a nanoparticle doping. The magnetic cobalt-ferrite nanoparticles localize in the vacant center of dislocation loops in the smectic textures and stabilize them.

By polarized microscopy in different geometries, the authors observe that nanoparticle clusters arise at the isotropic-smectic A phase transition and grow on cooling leading to persistent and spatially-regular textures in the Smectic A phase. A model based on smectic elasticity is proposed to explain this long-distance and thermally-reversible process. The experiments are well driven and the model accounting for the experimental results sounds well. The paper is well written and can be published as it is, only providing the two small comments listed below.

1- In section 3.1 and in Fig. 2-a can you precise the direction of the applied electric field.

2- In section 3.2 lines 189-190 :

As the clusters of nanoparticles have dimensions of micrometers, one can expect that they are structured into magnetically interacting aggregates. However, we expect ...

Reviewer 2 Report

The work is very weak.

The bibliography is poorly treated, not citing many other works.

The results presented do not rise to the level of an ISI paper.

Everything the authors discuss has been known for more than 2 decades. The article does not bring anything new.

I recommend sending it to a BDI journal.